# Clinical Characteristics of Mild Patients with Breakthrough Infection of Omicron Variant in China after Relaxing the Dynamic Zero COVID-19 Policy

**DOI:** 10.3390/vaccines11050968

**Published:** 2023-05-10

**Authors:** Yingyu He, Fang Zhang, Yan Liu, Zhou Xiong, Shangen Zheng, Wanbing Liu, Lei Liu

**Affiliations:** 1Hubei Key Laboratory of Central Nervous System Tumor and Intervention, Wuhan 430070, Chinawanbing1203x@163.com (W.L.); 2Department of Transfusion Medicine, General Hospital of Central Theater Command of the PLA, Wuhan 430070, China; 3Department of Anesthesiology, General Hospital of Central Theater Command of the PLA, Wuhan 430070, China

**Keywords:** COVID-19, omicron, breakthrough infection, booster vaccination, primary vaccination

## Abstract

For SARS-CoV-2 mutants, the effectiveness of the COVID-19 vaccines is still controversial. In this study, we aimed to investigate the clinical characteristics of Omicron-infected patients who completed primary immunization and booster immunization, respectively, during the rapid propagation of the Omicron variant in China. A total of 932 patients with confirmed SARS-CoV-2 infection from 18 December 2022 to 1 January 2023 were included in this survey by filling out questionnaires online. The enrolled patients were divided into the primary immunization group and the booster immunization group according to their vaccination status. During the whole course of disease, the most frequent symptoms were fever (90.6%), cough (84.3%), weakness (77.4%), headache and dizziness (76.1%), and myalgia (73.9%). Nearly 90% of the patients had symptoms lasting for less than 10 days, and 39.8% of the patients ended the course of the disease in 4–6 days. A total of 58.8% of these patients had a fever with a maximum body temperature of over 38.5 °C. Moreover, 61.4% of the patients had a fever that lasted less than 2 days. There were no obvious differences in initial symptoms, cardinal symptoms, symptom duration time, maximum body temperature, and fever duration time between the two groups of patients. In addition, no significant difference was found in the positive or negative conversion time of SARS-CoV-2 antigen/nucleic acid between the two groups of patients. For mild patients with Omicron breakthrough infection, enhanced immunization has no significant impact on the clinical performance and duration of viral infection compared with primary immunization. The reasons behind the different clinical manifestations of patients with mild symptoms after the breakthrough infection of the Omicron strain are still worth further research. Heterologous vaccination may be a better strategy for enhanced immunization, which can help improve the immune protection ability of the population. Further research should be carried out on vaccines against mutant strains and spectral anti-COVID-19 vaccines.

## 1. Introduction

Coronavirus disease 2019 (COVID-19) is caused by severe acute respiratory syndrome coronavirus 2 (SARS-CoV-2). SARS-CoV-2 has evolved, and new variants such as Alpha, Delta, and Omicron have emerged since the onset of COVID-19. The Omicron variant is currently widespread worldwide and has a high degree of mutation in infectivity and immune escape [1]. As of the end of April 2023, more than 680 million people worldwide have been infected with COVID-19, resulting in more than 6.8 million deaths. Vaccination is an important means of establishing an immune barrier. However, breakthrough infections in vaccinated populations, as well as repeat infections in previously infected populations, often occur due to the strong infectivity of Omicron and immune regression [2,3]. The protective effect of the COVID-19 vaccine against Omicron variant infection is limited and rapidly decreases over time [4,5,6,7,8], while the protective effect of the COVID-19 vaccine against severe illness or death after Omicron variant infection is significant [9,10,11,12]. Timely completion of enhanced immunization is an effective way to improve the immune protection ability to prevent the disease from developing into severe illness after infection. Nevertheless, it is still unknown whether the completion of enhanced immunization will affect the clinical performances of mild patients with Omicron breakthrough infection.

At the beginning of 2021, the Chinese government began to scientifically deploy and orderly promote the nationwide COVID-19 vaccination work. The main vaccines widely administered to the domestic population are three inactivated vaccines (made in Beijing or Wuhan Institute of Biological Products Co., Ltd., Beijing or Wuhan, China or Sinovac Life Sciences Co., Ltd., Beijing, China; respectively), adenovirus vector vaccines (made in CanSino Biologics Co., Ltd., Tianjin, China), and recombinant subunit vaccines (made in Anhui Zhifei Longcom Biopharmaceutical Co., Ltd., Hefei, China). Among them, the Chinese population receiving the three inactivated vaccines is the highest and most widespread. As of 2 March 2023, Chinese people have received a total of 3.49 billion vaccine doses, accounting for 90.58% of the total population. On 6 December 2022, China announced the cancellation of the dynamic zero COVID-19 policy after taking into account the epidemic characteristics of Omicron variant strains in China, the active immune level of the population, and the needs of social and economic development. Then, the Omicron variant strains quickly spread among the population. Up to now, it is estimated that nearly 80% of the population in China has been infected by Omicron, and the vast majority of infected people are cases of breakthrough infection that occur after vaccination and always show mild symptoms. However, we also found that the significant differences in the duration and manifestations of the disease, as well as the severity of the disease, existed even among these patients with mild infections. These different manifestations such as duration of virus infection may further have effects on the transmission of the Omicron variant. The duration and severity of illness can also have a significant impact on an individual’s work and life. Therefore, we want to clarify whether completing the booster vaccination will have a significant impact on the clinical manifestations of these mild patients. Additionally, immunological memory extinction is a common phenomenon after natural immunity or artificial immunity, so we further analyzed the impact of time intervals for strengthening immunization.

In this study, we made a comparative analysis about the clinical characteristics between the mild patients with Omicron infection who completed primary immunization and booster immunization, respectively, during the rapid spread of Omicron in China and after relaxing the dynamic zero COVID-19 policy. We hope that our research results can provide some basis for the prevention and control policies of COVID-19 and COVID-19 vaccination policies in the next stage.

## 2. Materials and Methods

### 2.1. Study Design and Participants

This investigation was performed between 18 December 2022 and 1 January 2023 (during the rapid spread of Omicron in China). All participants who were infected with Omicron and the course of disease ended were recruited online and completed an online self-administered questionnaire through Wenjuanxing. The general information and clinical data of 1551 patients were collected in total. Of these, 619 cases were excluded (403 patients had no results for SARS-CoV-2 nucleic acid or antigen test during the course of disease, 71 patients had incomplete information, 69 patients failed to vaccinate or completed primary immunization, the inoculated vaccines of 53 patients were not inactivated vaccines, 16 patients were hospitalized, and the permanent residence of 7 patients was not in China). The remaining 932 patients were enrolled in this study (Figure 1). These patients all showed mild symptoms after being infected with Omicron and at least completed the primary immunization of COVID-19 inactivated vaccines (made in Beijing or Wuhan Institute of Biological Products Co., Ltd., Beijing or Wuhan, China or Sinovac Life Sciences Co., Ltd., Beijing, China). Completing primary immunization means receiving two doses of inactivated vaccines. Completing booster immunization means receiving three doses of inactivated vaccines. That is to say, these enrolled research subjects all used homologous enhanced immunization. The vaccination dose of each dose of inactivated vaccine is in accordance with the standards formulated by the health departments in China. The time interval between booster immunization and primary immunization is generally more than six months. According to whether they had finished booster immunization or not, the enrolled patients were divided into two groups (the primary immunization group and the booster immunization group). Moreover, the patients in the booster immunization group were further divided into two groups (the less than 1 year group and the more than 1 year group) according to whether the time interval between booster vaccination and breakthrough infection was more than one year.

### 2.2. Data Collection

A self-designed structured questionnaire was used to obtain information about the enrolled subjects. The general information including age, sex, occupation, coexisting disorders, past history of COVID-19, medical treatment, used drugs, and the most likely infection channels were obtained. The clinical manifestations including typical clinical symptoms, initial symptoms, symptom duration time, maximum body temperature, fever duration time, days of antigen/nucleic acid positive after symptom onset, and days of antigen/nucleic acid negative after symptom onset were carefully recorded. Some of the patients only underwent SARS-CoV-2 nucleic acid or antigen testing during the course of the disease, while others underwent both tests. For the patients who underwent both SARS-CoV-2 nucleic acid and antigen testing, the days of antigen/nucleic acid positive after symptom onset were calculated as the time when antigen or nucleic acid first appeared positive. The days of antigen/nucleic acid negative after symptom onset were calculated as the time when antigen or nucleic acid last appeared negative.

### 2.3. Statistical Analyses

Classification variables are expressed as numbers/percentages. Variables were compared using the chi-square test, while Fisher’s exact test was used when data were limited. SPSS 26.0 software was used for statistical analysis. A two-sided test of α less than 0.05 was considered statistically significant.

## 3. Results

### 3.1. Demographics and Characteristics of Patients Infected with SARS-CoV-2 Omicron Variant

A total of 932 subjects were enrolled in the cohort. Among them, 537 cases completed booster immunization, while 395 cases just completed primary immunization. The general characteristics of these patients were shown in Table 1. There were no significant differences both in age and sex between the two groups. The proportion who completed enhanced immunization was higher than that who completed basic immunization among teachers, which was the opposite to the case among company staff (*p* < 0.05). Additionally, no significant differences were found in combined basic diseases and past history of COVID-19 between the two groups of patients. In total, 80.8% of the patients treated diseases through self-purchased medicines. Antipyretic and analgesic drugs were the most frequently used drugs, accounting for 84.2%. In addition, 31.7% of the patients chose traditional Chinese medicine treatment. Regarding the most likely infection mode, 29.7%, 27.6%, and 22.2% of the patients were presumed to be infected by co-workers, family members, and strangers in public places, respectively.

### 3.2. Clinical Manifestations of Patients Infected with SARS-CoV-2 Omicron Variant

The clinical performances of the two groups of patients were shown in Table 2. The most common initial symptoms were fever (36.2%), pharyngalgia (23.2%), and myalgia (11.9%). During the whole course of disease, the most frequent symptoms were fever (90.6%), cough (84.3%), weakness (77.4%), headache and dizziness (76.1%), and myalgia (73.9%). There were no obvious differences in cardinal symptoms except for pharyngalgia, ageusia, and nausea between the two groups of patients. Nearly 90% of the patients had symptoms lasting for less than 10 days, and 39.8% of the patients ended the course of the disease in 4–6 days. A total of 58.8% of these patients had a fever with a maximum body temperature of over 38.5 °C. Moreover, 61.4% of the patients had a fever that lasted less than 2 days. However, there were no significant differences in symptom duration time, maximum body temperature, and fever duration time between the two groups of patients. The antigen/nucleic acid results of most patients (74.6%) turned positive within 2 days of the onset of the disease. Additionally, the antigen/nucleic acid results of 87.3% of the patients turned negative within 10 days of the onset of the disease.

### 3.3. Clinical Manifestations of Patients with Different Booster Immunization Time Interval

Among the patients who completed booster immunization, the time interval between vaccination and breakthrough infection of 319 cases was less than one year, and that of the other 218 cases was more than one year. We observed no significant differences in both initial symptoms and cardinal symptoms during the whole illness between the two groups of patients (Table 3). Similarly, no obvious differences were found in symptom duration time, maximum body temperature, and fever duration time between the two groups of patients. In addition, the time for antigen/nucleic acid to turn positive or negative also did not show significant differences.

## 4. Discussion

The wide spread of the epidemic is found to be closely related to the continuous evolution and variation of SARS-CoV-2 since the beginning of COVID-19 [13,14], which brings great challenges to epidemic prevention and control globally. A great deal of evidence indicates that COVID-19 vaccination against the original strain can reduce the number of infected cases and the severity of the disease [15,16,17,18]. However, for mutants, the effectiveness of the vaccination is still controversial [19,20,21]. The mutants, especially the Omicron variant, dramatically increase the transmissibility and the ability to escape from natural and vaccine-induced immunity. This raises concerns about the ability of the original COVID-19 vaccination to prevent Omicron infection. The latest studies show that vaccination, especially inoculating booster doses, and previous infections still have a certain role in controlling new infections during the pandemic of the Omicron variant [22,23]. In the context of the global pandemic of the Omicron strain, many people who have been vaccinated or previously infected with COVID-19 still have breakthrough infections. These people with breakthrough infection mainly show mild symptoms such as fever, cough, fatigue, sore throat, etc. This is mainly due to the reduction of the virulence of the Omicron strain and the existence of a certain degree of immune response and immunological memory in the population. However, as mild patients, there are still significant differences in the symptoms and duration of these infected individuals. Some patients even have long COVID symptoms. It is still unknown whether these differentiated performances are related to the patients’ immune response status and level.

In this study, we plan to analyze the impact of enhanced immunization on the clinical performance and disease transmission ability in the mild patients with breakthrough infection of the Omicron variant. On 6 December 2022, the Omicron variant began to spread quickly among community populations in China after relaxing the dynamic zero COVID-19 policy. We found that the vast majority of patients could cure their diseases by purchasing drugs themselves and that the course of the disease ended quickly. The vast majority of patients had a disease course of about 7 days, usually no more than 10 days. This result further confirms that the Omicron variant is extremely infectious and weakly pathogenic and that original vaccination is likely to still have an effect on severe prevention. More than 90% of these patients experienced fever symptoms, but the duration of the fever was generally not long (mostly not exceeding 3 days). Nevertheless, over half of the patients had a maximum body temperature above 38.5 °C. Effectively controlling body temperature is a concern to prevent sustained high fever from causing other complications. Most patients experienced a positive conversion of SARS-CoV-2 antigen or nucleic acid within two days of symptom onset, and a negative conversion occurs around 7 days after symptom onset. Overall, mild patients often have a duration of virus infection of no more than one week. However, we also found that there were no significant impacts on symptom manifestations and duration of virus infection (according to the time of antigen/nucleic acid positive/negative after symptom onset) in people who had completed enhanced immunization compared to those who had only completed primary immunization. We consider that the reasons may be that the long time between strengthening immunization and breakthrough infection leads to obvious immune regression. This is due to a rapid decrease in vaccine-induced peak IgA, which is a mucosal antibody with a stronger neutralizing activity against spike protein than IgG [24,25].

Several studies have shown that the protective effect of booster immunization on severe COVID-19 can be maintained steadily for a year, while the protection against symptomatic infection decreased significantly over time after the last vaccination [26,27,28]. Therefore, we further grouped according to the time interval for enhanced immunization. Nevertheless, we also found no significant differences in the symptoms’ manifestations and the duration of virus infection between patients receiving enhanced vaccination within one year and more than one year. We analyze the reasons behind this, which may be that the rapid decline in immune response induced by enhanced immunization is insufficient to effectively prevent Omicron variant infection and remove the virus. Vaccine-induced protection against SARS-CoV-2 infection is temporary and is substantially affected by evolution and variation of the virus. Although the primary purpose of vaccination is to protect individuals from severe COVID-19 and its consequences, the extent to which vaccines can reduce the continued spread of COVID-19 is key to containing the epidemic. It depends on the ability of the vaccine to prevent infection and the extent to which vaccination reduces the infectiousness of breakthrough infections. Considering the diminished effectiveness of the vaccine against infection of SARS-CoV-2 new mutant strains, we should establish a scientific enhanced immunization cycle to provide more extensive protection. The selection of enhanced immunization strategies is also an important issue. Heterologous enhanced immunization usually achieves a stronger level of immune response compared to homologous enhanced immunization. Based on the fact that the vaccines administered to Chinese citizens in the early stage are mainly inactivated vaccines, other vaccines such as adenovirus vector vaccines and recombinant subunit vaccines can be chosen for future booster immunization. In addition, it is also necessary to conduct targeted vaccine research on new mutant strains or more spectral vaccine research. Despite the significant benefits in terms of reduced risk of hospitalization and death, different adverse events such as pain, soreness, itching at the vaccination site, and fever, headache, and muscle soreness may present after vaccination. A systematic literature review and meta-analysis demonstrates that vaccines are associated to a two-fold risk of developing a headache within 7 days of injection, and the lack of difference between vaccine types enables us to hypothesize that headache is secondary to systemic immunological reaction rather than to a vaccine-type specific reaction [29]. However, the mechanism of headache after vaccination is not clear but may also be related to dangerous complications.

## 5. Conclusions

For people who exhibit mild symptoms after breakthrough infection with Omicron, the homologous enhanced immunization using inactivated vaccines seems to have no significant impact on the symptom manifestations, duration, severity, and duration of virus infection compared to primary immunization. Therefore, using allogeneic immunization for enhanced vaccination may be a better option. We should strengthen research on vaccines against mutant strains and spectral anti-COVID-19 vaccines.

## Figures and Tables

**Figure 1 vaccines-11-00968-f001:**
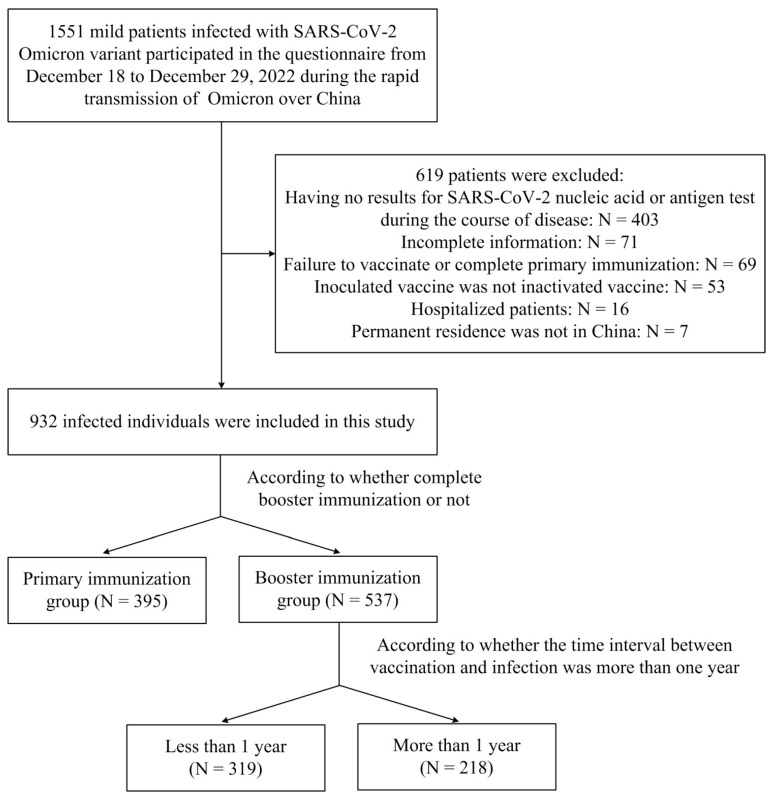
Study population.

**Table 1 vaccines-11-00968-t001:** Demographics and characteristics of patients infected with SARS-CoV-2 Omicron variant.

Characteristics	All Patients (N = 932)	Primary Immunization (N = 395)	Booster Immunization (N = 537)	*p* Value
Age groups (years)				
<14	5 (0.5)	5 (1.2)	0 (0.0)	0.088
14–29	392 (42.1)	165 (41.8)	227 (42.3)	
30–45	442 (47.4)	188 (47.6)	254 (47.3)	
46–59	84 (9.0)	32 (8.1)	52 (9.7)	
>59	9 (1.0)	5 (1.3)	4 (0.7)	
Sex				
Male	362 (38.8)	141 (35.7)	221 (41.2)	0.091
Female	570 (61.2)	254 (64.3)	316 (58.8)	
Occupation				
Teacher **	49 (5.3)	11 (2.8)	38 (7.1)	0.019 *
Student	110 (11.8)	46 (11.6)	64 (11.9)	
Self-employed	44 (4.7)	22 (5.6)	22 (4.0)	
Company Staff **	216 (23.2)	106 (26.8)	110 (20.5)	
Public servants	47 (5.0)	17 (4.3)	30 (5.6)	
Healthcare workers	201 (21.6)	89 (22.5)	112 (20.9)	
Others	265 (28.4)	104 (26.4)	161 (30.0)	
Coexisting disorders				
Hypertension	39 (4.2)	21 (5.3)	18 (3.4)	0.139
Diabetes	13 (1.4)	7 (1.8)	6 (1.1)	0.400
Respiratory diseases	15 (1.6)	8 (2.8)	7 (1.3)	0.387
Cardiovascular and cerebrovascular diseases	9 (1.0)	5 (1.3)	4 (0.7)	0.422
Tumor	5 (0.5)	4 (1.0)	1 (0.2)	0.088
Past history of COVID-19				
Yes	21 (2.3)	12 (3.0)	9 (1.7)	0.166
No	911 (97.7)	383 (97.0)	528 (98.3)	
Medical treatment				
Untreated	141 (15.1)	57 (14.4)	84 (15.6)	0.871
Self-purchased medicine	753 (80.8)	322 (81.5)	431 (80.3)	
Fever outpatient treatment	38 (4.1)	16 (4.1)	22 (4.1)	
Drugs use				
Antipyretics and analgesics	666 (84.2)	282 (83.4)	384 (84.8)	0.610
Antiviral drugs	202 (25.5)	80 (23.7)	122 (26.9)	0.298
Antibiotics	127 (16.1)	55 (16.3)	72 (15.9)	0.886
Chinese medicine	251 (31.7)	81 (24.0)	170 (37.5)	<0.001*
Fluid administration	15 (1.9)	5 (1.5)	10 (2.2)	0.458
Most likely infection mode				
Family	257 (27.6)	107 (27.1)	150 (27.9)	<0.001 *
Colleagues **	277 (29.7)	102 (25.8)	175 (32.6)	
Confronting and treating patients	174 (18.7)	70 (17.7)	104 (19.4)	
Public places **	207 (22.2)	109 (27.6)	98 (18.2)	
Takeaway delivery	17 (1.8)	7 (1.8)	10 (1.9)	

* For overall comparison, *p* < 0.05. ** For individual item comparison, *p* < 0.05.

**Table 2 vaccines-11-00968-t002:** Clinical manifestations of patients infected with SARS-CoV-2 Omicron variant.

Characteristics	All Patients (N = 932)	Primary Immunization (N = 395)	Booster Immunization (N = 537)	*p* Value
Initial symptom				
Fever	337 (36.2)	129 (32.7)	208 (38.7)	0.160
Pharyngalgia	216 (23.2)	85 (21.4)	131 (24.4)	0.304
Myalgia	111 (11.9)	53 (13.4)	58 (10.8)	0.223
Headache and dizziness	92 (9.8)	47 (11.9)	45 (8.4)	0.075
Cough and expectoration	65 (7.0)	37 (9.4)	28 (5.2)	0.014 *
Cardinal symptoms				
Fever	844 (90.6)	355 (89.9)	489 (91.1)	0.540
Cough	786 (84.3)	336 (85.1)	450 (83.8)	0.600
Weakness	721 (77.4)	297 (75.2)	424 (79.0)	0.174
Dizziness and headache	709 (76.1)	300 (75..9)	409 (76.2)	0.940
Myalgia	689 (73.9)	291 (73.7)	398 (74.1)	0.897
Pharyngalgia	647 (69.4)	256 (64.8)	391 (72.8)	0.009 *
Expectoration	626 (67.2)	264 (66.8)	362 (67.4)	0.853
Poor appetite	494 (53.0)	215 (54.4)	279 (52.0)	0.454
Hoarseness	412 (44.2)	172 (43.5)	240 (44.7)	0.727
Ageusia	328 (35.2)	162 (41.0)	166 (30.9)	0.001 *
Anosmia	276 (29.6)	123 (31.1)	153 (28.5)	0.382
Nausea	252 (27.0)	92 (23.3)	160 (29.8)	0.027 *
Diarrhea	205 (22.0)	82 (20.8)	123 (22.9)	0.435
Vomiting	172 (18.5)	72 (18.2)	100 (18.6)	0.878
Dyspnea	116 (12.4)	57 (14.4)	59 (11.0)	0.116
Symptom duration (d)				
1–3	216 (23.3)	100 (25.3)	116 (21.6)	0.150
4–6	372 (39.8)	141 (35.7)	231 (43.0)	
7–10	245 (26.2)	105 (26.6)	140 (26.1)	
11–14	49 (5.3)	23 (5.8)	26 (4.8)	
≥15	50 (5.4)	26 (6.6)	24 (4.5)	
Maximum body temperature (°C)				
<37.0	49 (5.3)	18 (4.6)	31 (5.8)	0.202
37.0–38.5	335 (35.9)	150 (38.0)	185 (34.5)	
38.6–39.9	520 (55.8)	211 (53.4)	309 (57.5)	
≥40.0	28 (3.0)	16 (4.0)	12 (2.2)	
Fever duration (d)				
0–2	542 (61.4)	242 (64.2)	300 (59.3)	0.500
3–4	310 (35.1)	122 (32.4)	188 (37.2)	
5–6	28 (3.2)	12 (3.2)	16 (3.2)	
≥7	3 (0.3)	1 (0.2)	2 (0.3)	
Days of antigen/nucleic acid positive after symptom onset (d)				
0	123 (13.2)	40 (10.1)	83 (15.5)	0.191
1	287 (30.8)	124 (31.4)	163 (30.4)	
2	285 (30.6)	128 (32.4)	157 (29.2)	
3	150 (16.1)	68 (17.2)	82 (15.3)	
4	38 (4.1)	13 (3.3)	25 (4.7)	
≥5	49 (5.3)	22 (5.6)	27 (5.0)	
Days of antigen/nucleic acid negative after symptom onset (d)				
1–3	65 (7.0)	26 (6.6)	39 (7.3)	0.974
4–6	211 (22.6)	87 (22.0)	124 (23.1)	
7–10	538 (57.7)	230 (58.2)	308 (57.4)	
11–14	78 (8.4)	35 (8.9)	43 (8.0)	
≥15	40 (4.3)	17 (4.3)	23 (4.3)	

* For overall comparison, *p* < 0.05.

**Table 3 vaccines-11-00968-t003:** Clinical manifestations of patients with different booster immunization time interval.

Characteristics	All Patients (N = 537)	≤1 Year (N = 319)	>1 Year (N = 218)	*p* Value
Initial symptom				
Fever	209 (38.9)	121 (37.9)	88 (40.4)	0.570
Pharyngalgia	131 (24.4)	75 (23.5)	56 (25.7)	0.564
Myalgia	58 (10.8)	40 (12.5)	18 (8.3)	0.116
Headache and dizziness	45 (8.4)	25 (7.8)	20 (9.2)	0.583
Cough and expectoration	28 (5.2)	15 (4.8)	13 (6.0)	0.519
Cardinal symptoms				
Fever	489 (91.1)	290 (90.9)	199 (91.3)	0.881
Cough	450 (83.8)	272 (85.3)	178 (81.7)	0.264
Weakness	424 (79.0)	256 (80.3)	168 (77.1)	0.374
Dizziness and headache	409 (76.2)	244 (76.5)	165 (75.7)	0.831
Myalgia	398 (74.1)	231 (72.4)	167 (76.6)	0.276
Pharyngalgia	391 (72.8)	235 (73.7)	156 (71.6)	0.590
Expectoration	362 (67.4)	221 (69.3)	141 (64.7)	0.264
Poor appetite	279 (52.0)	169 (53.0)	110 (50.5)	0.566
Hoarseness	240 (44.7)	144 (45.1)	96 (44.0)	0.800
Ageusia	166 (30.9)	105 (32.9)	61 (28.0)	0.224
Anosmia	153 (28.5)	101 (31.7)	52 (23.9)	0.050
Nausea	160 (29.8)	101 (31.7)	59 (27.1)	0.253
Diarrhea	123 (22.9)	75 (23.5)	48 (22.0)	0.686
Vomiting	100 (18.6)	62 (19.4)	38 (17.4)	0.558
Dyspnea	59 (11.0)	38 (11.9)	21 (9.6)	0.407
Symptom duration (d)				
1–3	116 (21.6)	72 (22.6)	44 (20.2)	0.728
4–6	231 (43.0)	140 (43.9)	91 (41.7)	
7–10	140 (26.1)	79 (24.8)	61 (28.0)	
11–14	26 (4.8)	13 (4.0)	13 (6.0)	
≥15	24 (4.5)	15 (4.7)	9 (4.1)	
Maximum body temperature (°C)				
<37.0	31 (5.8)	20 (6.3)	11 (5.0)	0.762
37.0–38.5	185 (34.5)	113 (35.4)	72 (33.0)	
38.6–39.9	309 (57.5)	180 (56.4)	129 (59.2)	
≥40.0	12 (2.2)	6 (1.9)	6 (2.8)	
Fever duration (d)				
0–2	300 (59.2)	173 (57.9)	127 (61.4)	0.771
3–4	188 (37.2)	114 (38.1)	74 (35.7)	
5–6	16 (3.2)	11 (3.7)	5 (2.4)	
≥7	2 (0.4)	1 (0.3)	1 (0.5)	
Days of antigen/nucleic acid positive after symptom onset (d)				
0	123 (13.2)	40 (10.1)	83 (15.5)	0.410
1	287 (30.8)	124 (31.4)	163 (30.4)	
2	285 (30.6)	128 (32.4)	157 (29.2)	
3	150 (16.1)	68 (17.2)	82 (15.3)	
4	38 (4.1)	13 (3.3)	25 (4.7)	
≥5	49 (5.3)	22 (5.6)	27 (5.0)	
Days of antigen/nucleic acid negative after symptom onset (d)				
1–3	44 (8.2)	28 (8.8)	16 (7.3)	0.162
4–6	119 (22.2)	75 (23.5)	44 (20.2)	
7–10	305 (56.8)	183 (57.4)	122 (56.0)	
11–14	41 (7.6)	17 (5.3)	24 (11.0)	
≥15	28 (5.2)	16 (5.0)	12 (5.5)	

## Data Availability

The data that support the findings of this study are available from the corresponding author upon reasonable request.

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
