# Peer review of "Clinical Characteristics of Mild Patients with Breakthrough Infection of Omicron Variant in China after Relaxing the Dynamic Zero COVID-19 Policy"

_vaccines, 2023, doi:10.3390/vaccines11050968_

Round 1

Reviewer 1 Report

In this paper authors compared the severity of Omicron among two group of patients: (1) primary immunization group, and (2) booster immunization group. They did not find any significant difference between the two groups based on the clinical consequences. 

In my opinion this study is missing one major control group, i.e. control patients who are not vaccinated.

While reading this article few questions came to my mind.

What scientific question asked in this study is not clear.

Since the authors presenting a negative outcome, how they prove that the intervention model is a good model to study.

In this study there is no difference between primary immunization and booster immunization group. What does it mean? People should not take booster immunization?

It looks like the study design is not appropriate. 

Reviewer 2 Report

The paper presented to me for review deals with the practically important issue of "clinical characteristics of mild patients with breakthrough infection of omicron variant in china after relaxing the dynamic zero COVID-19 policy" and addresses the problems of vaccine efficacy against SARS-CoV-2 mutants. The work is based on a very large group of patients and so the statistics are reliable. The work contributes new knowledge in the topic, it is written according to the typical scheme, the results and conclusions are logical

However, before accepting the paper for publication, I would complete one point:

1. the authors showed that one of the most common adverse effect after vaccination was headache, a huge meta-analysis: https://pubmed.ncbi.nlm.nih.gov/35361131/ should be referred to in the discussion and included in the references emphasizing that still the mechanism of headache after vaccination is not clear but may also be related to dangerous complications 

Reviewer 3 Report

This study presents the results of a symptom questionnaire for totally 932 patients with mild covid-19 infections. The authors did not find any major difference in symptoms with those who had primary immunization vs. those who had had a booster dose. As such, the study is appropriate even though with little scientific merit. There are, however, some issues with the methodology:

- there is actually no information provided concerning the vaccines that the study subjects received

- there are multiple comparisons in the statistical analyses but no adjustment of p-values

- the tables are extensive, but they do not present well the key findings 

Round 2

Reviewer 1 Report

The overall writing is better than before